# Function of the Long Noncoding RNAs in Hepatocellular Carcinoma: Classification, Molecular Mechanisms, and Significant Therapeutic Potentials

**DOI:** 10.3390/bioengineering9080406

**Published:** 2022-08-21

**Authors:** Ahmad Khan, Xiaobo Zhang

**Affiliations:** College of Life Sciences, Zhejiang University, Hangzhou 310058, China

**Keywords:** hepatocellular carcinoma, lncRNAs, mechanisms, biomarker

## Abstract

Hepatocellular carcinoma (HCC) is the most common and serious type of primary liver cancer. HCC patients have a high death rate and poor prognosis due to the lack of clear signs and inadequate treatment interventions. However, the molecular pathways that underpin HCC pathogenesis remain unclear. Long non-coding RNAs (lncRNAs), a new type of RNAs, have been found to play important roles in HCC. LncRNAs have the ability to influence gene expression and protein activity. Dysregulation of lncRNAs has been linked to a growing number of liver disorders, including HCC. As a result, improved understanding of lncRNAs could lead to new insights into HCC etiology, as well as new approaches for the early detection and treatment of HCC. The latest results with respect to the role of lncRNAs in controlling multiple pathways of HCC were summarized in this study. The processes by which lncRNAs influence HCC advancement by interacting with chromatin, RNAs, and proteins at the epigenetic, transcriptional, and post-transcriptional levels were examined. This critical review also highlights recent breakthroughs in lncRNA signaling pathways in HCC progression, shedding light on the potential applications of lncRNAs for HCC diagnosis and therapy.

## 1. Introduction

Cancer, a major public health issue and one of the world’s lethal illnesses [1,2,3,4,5,6], is a multifaceted disorder characterized by uncontrolled cellular growth through genetic variations, epigenetic changes, chromosomal rearrangements, and amplification [7,8,9]. Various cancers have been linked to increased causes of death and mortality, despite the best efforts of experts to conduct comprehensive investigations to develop more effective therapeutic techniques [10,11]. As a consequence, finding effective screening tools, diagnostic biomarkers, and more effective treatment approaches to increase tumor patients’ long-term survival and treatment rates is essential [12,13,14].

Hepatocellular carcinoma (HCC), a severe specific type of primary liver cancer [15], accounts for 75–85% of cases of death [16,17,18,19,20]. The low survival rate of HCC is due to asymptomatic initiation in premature stages and loss of optimum treatment period after diagnosis in middle or late stages [17]. Hepatitis B or C virus infection, aflatoxin B1, drugs, alcohol, and metabolic diseases are the most common risk factors for HCC [21,22,23,24,25]. Moreover, there are a few uncommon diseases that increase HCC risks, including alpha1-antitrypsin deficiency (A1AT), tyrosinemia, and Wilson disease. Tyrosinemia is caused by deficiency of fumarylacetoacetate hydrolase, whereas Wilson’s disease is caused by a mutation in the ATP7B gene. All of these are complex multifactorial diseases that cause hepatotoxicity via various mechanisms, which can eventually lead to cirrhosis and HCC [24].

The initiation and persistence of HCC are complicated and are influenced by multiple variables [26,27]. HCC is associated with high levels of tumor growth, postoperative relapse, and chemo resistance [28,29,30,31,32]. Hepatic fibrosis, a wound-healing condition that involves the dysregulation of extracellular matrix proteins and alteration of normal hepatic architecture, is a significant risk factor for HCC [33,34,35]. The regulatory mechanisms involved in HCC are still hot topics [36,37]. HCC progression is a complex mediated through accumulation of genetic and epigenetic modifications [38,39], accumulating the necessary amount of genetic and epigenetic variations, leading to the formation of dysplastic foci and nodules, eventually progressing into HCC (Figure 1). To elucidate the HCC progression, landscape, and biology, a complete transcriptome study of the specimens demonstrating diverse disease stages may offer a higher-resolution view of the essential mechanisms of progression [40]. Nonalcoholic fatty liver disease (NAFLD) has been increasing in prevalence and is defined as excessive fat accumulation in the liver and steatosis presence in >5% of hepatocytes [41]. NAFLD can develop to nonalcoholic steatohepatitis (NASH), with inflammation and ballooning with or without fibrosis. NASH further develops to liver cirrhosis in a significant proportion of the patients and eventually progresses into HCC [41]. Generally, NAFLD HCC patients have poor prognosis and are associated with a more progressive stage of the disease [42]. Improved control of virus C illnesses and latent liver cancer has resulted in an increasing number of patients with restored liver function in response to cirrhosis [43]. Systemic chemotherapy, molecule-targeted therapy, trans catheter artery chemoembolization, and immunotherapy are the most common and effective treatments [27].

To formulate new diagnostic and therapeutic approaches against HCC and to enhance the prognostic value of diagnosed patients, it is important to reveal the relationship among signs, symptoms, and molecular alterations [44]. Advances in biomedical technology to date, such as live transplantation, surgical excision, and radiofrequency ablation, has increased the 5-year survival rates of HCC patients [45,46,47]. Additional molecular mechanisms and the development of reliable biological indicators for HCC detection are critical at the initial stage of HCC development [26,48,49,50,51,52].

Carcinogenesis is frequently caused by abnormal expressions of genes [53,54]. Recent evidence suggests that RNA processing has been consistently changed in cancer [55,56,57], revealing the critical role of RNA in tumor genesis and cancer development [54,58]. The long noncoding RNAs (lncRNAs) are mainly categorized according to their positional relationship with adjacent coding genes [59]. Several reports have been published in the scientific literature, highlighting a potential role for lncRNAs in tissue pathophysiology and development [20,49,60,61]. Evidence has shown that lncRNAs are mostly dysregulated as tumor suppressors in various cancers [20,62], and many lncRNAs are intricately linked to the progression of cancer, including HCC [25,63,64,65,66,67], signifying that lncRNAs are potential therapeutic targets in HCC [68].

In this review article, we present an overview of the existing knowledge on lncRNAs in HCC progression and analyze their mechanisms in the cancer phenotype. We also discuss the prospective application of lncRNAs as prognostic and therapeutic targets for HCC patients with future prospective to recognize diverse mechanisms of lncRNAs in HCC.

## 2. Characteristics and Classification of RNAs

RNA sequencing technology has identified more than one hundred thousand (100,000) distinct RNA molecules of mammalian species [69,70,71]. Coding RNAs and noncoding RNAs (ncRNAs) are the two types of RNAs [72,73]. Based on the length of transcripts, ncRNAs can be divided into two classes (small ncRNAs and long ncRNAs). The miRNAs (microRNAs), snRNAs (small nucleolar RNAs), PIWI-interacting RNAs, and other endogenous RNAs are examples of small ncRNAs [53,54], which have a nucleotide number of less than 200 [74]. LncRNAs (long noncoding RNAs), lincRNAs (long intergenic noncoding RNAs), NATs (natural antisense transcripts), T-UCRs (transcribed ultra-conserved regions), long enhancer ncRNAs, and noncoding repeat sequences, as pseudo genes, are examples of long ncRNAs with more than 200 nucleotides (Figure 2) [74].

Small ncRNAs were first identified by exogenous RNA interference (RNAi) in plants and nematodes and were found to exist endogenously, functioning mostly as gene regulators through pairing to the target genes, hence directing their post-transcriptional activities in animals and plants [75]. It is well known that ncRNAs account for the majority of the human transcriptome, including miRNAs, lncRNAs, and circRNAs. MicroRNAs are single-stranded RNAs and participate in a series of physiological and pathological processes by facilitating post-transcriptional regulation of the target genes [76]. Numerous abnormally expressing miRNAs are associated with HCC initiation and progression [76,77]. Various studies have exposed the biological roles of lncRNAs as regulators of transcription, modulators of mRNA processing, and organizers of nuclear domains [76,77,78,79]. Compared with linear RNAs, circRNAs are more stable to exonuclease and ribonuclease, with conserved structure and stable sequence and tissue specificity [78,79]. It has been shown that circRNAs play significant pathophysiological roles in the existence and development of alcoholic liver injury; hepatic fibrosis, HCC, and other liver diseases CircRNAs have also been confirmed to exert effects with respect to regulation of cellular metabolisms of HCC [78].

For example, small ncRNAs, siRNAs, and/or miRNAs, have been well characterized [74,80,81]. LncRNAs, in comparison with small ncRNAs, are less understood in terms of their mechanisms and functions [74]. The prevalence of various forms of RNAs is altered in most eukaryotic cells. Ribosomal RNAs are responsible for approximately 80–85% of cellular RNA mass, accompanied mostly by tRNAs and mRNAs [82]. Although ncRNAs are not translated into proteins, they play important roles in the physiological functions of organisms [69,83]. In particular, lncRNAs are essential controllers of chromatin dynamics, growth, differentiation, and gene development [20,84]. At present, with the advancement of high-throughput sequencing and DNA tiling array technology, a number of investigations are concentrating on ncRNAs [85,86,87]. The functions of ncRNA-encoding peptides and proteins have prospective applications in cancers, with some potential challenges [88].

## 3. Characteristics and Functions of lncRNAs

Relying on the genetic position concerning neighboring protein-coding genomes, lncRNAs have been classified into five categories (Figure 2) [89,90]. The first category is the sense lncRNAs, which interact with protein coding gene. The specific genes on the sense strand are transcribed from the sense strand of the genome concerning protein-coding genes, such as COLDAIR [9,91]. The second group of lncRNAs is the antisense lncRNAs, which are transcribed from the antisense strand of the genome, such as lncRNA ANRIL. These lncRNAs interact with one and sometimes most exons of the protein-coding genome upon its reverse strand [9,91]. The third category is the bidirectional lncRNAs; for example, the lncRNA-enhancing eNOS (endothelial nitric oxide synthase) expression (LEENE) and lncRNA HCCL5, in this category the lncRNA and a protein-coding gene are located on the opposite sides of the genome and are derived from different directions of protein-coding genes [9,17,91]. The fourth intronic lncRNAs are generated entirely within the introns of the protein-coding genes, with no exons overlapping [9,91]. The fifth group of intergenic lncRNAs is found nearby almost no protein-coding genes [9,17,91,92,93,94].

LncRNAs can also be classified according to their targeting mechanisms, including signal, decoy, and scaffolds [9,95]. The lncRNA signal can control cell-specific expression in response to numerous stimuli [9,96]. Some lncRNAs function as decoys to negatively regulate target expression, acting as a molecular basin to dilute the cellular level of protein or other miRNAs [95,97]. Some lncRNAs act as scaffolds to a prearranged telomerase complex by accumulating modular binding sites for telomeric regulatory proteins [17,95]. Many investigations have found that lncRNAs mainly interact with miRNAs to execute their biological functions as competing endogenous RNAs (ceRNAs) [28,98,99]. In turn, miRNAs may directly interact with lncRNAs to silence their expressions. Various lncRNAs are difficult to classify in specific classifications [20,28,49,98,99].

The amount of illustrated lncRNAs has changed dramatically in recent years due to sequencing technologies. More than 50,000 lncRNAs have been described, with almost 58,000 lncRNA transcripts assembled in the Encyclopaedia of DNA Elements (ENCODE), and Project Consortium (GENCODE release 36), with 27,919 lncRNAs of humans and the elevated 50 ending in the Functional Annotation of Mammalian Genome (FANTOM5) [25]. There may be more than 15,000 lncRNAs in the human genome. Their expression is highly regulated by transcription factors and methylated lysines, such as mRNAs [100]. A more specific definition of lncRNA is an RNA molecule that cannot code for proteins and has a length of 200 bp to 100 kbp [9,54]. LncRNAs can have an open reading frame of more than 100 amino acids [101]. Polypeptides with fewer than 100 amino acids can be useful in species and are not considered byproducts of authoritative proteins [101]. RNA polymerase II transcribes the largest portion of lncRNAs, which is most often capped and polyadenylated [102], unlike mRNAs, which are highly conserved between humans and rodents [103].

LncRNAs have species- and tissue-specific expression patterns, which may relate to their key roles [73,103]. The three fundamental levels of lncRNA structure and sequence composition are primary, secondary, and tertiary [104]. The structural properties of lncRNAs assist researchers to improve their understanding of the chemical mechanisms that enable lncRNAs to perform their roles. Secondary structures of lncRNAs typically include duplexes, internal loops, junctions, and bulges, which can serve as protein-binding sites and are important components of operational lncRNAs, such as Watson-Crick complementary base pairing and stability of unpaired locations [105,106]. Terminal differentiation-induced noncoding RNA is conserved at its 5′ ends across vertebrates other than mice [107], but the 3′ end indicates the difference in sequence in vertebrates [73,108]. The triple helix at the 3′ ends of lncRNAs can stabilize the poly (A) tail-lacking lncRNAs. It also contributes to the structure of lncRNAs by providing interactive interfaces and preserving lncRNA stabilization [29,109].

LncRNAs have been found to play important roles not only in the normal biological functions of cells but also in the pathophysiological behaviors of various illnesses. Particularly tumors, through chromosome alteration, splicing, transcription factor activation, mRNA fragmentation, and other mechanisms (Figure 3) [25,29,110,111,112,113,114,115,116]. LncRNAs are considered to have significant regulative functions in pathogenesis with respect to the development of various human diseases. Proliferation, apoptosis, differentiation, and tumor growth are only a few examples that describe the functions of lncRNAs [117,118,119]. LncRNAs are sometimes expressed abnormally in tumors [120]. They can function as oncogenes or tumor suppressor drivers [88,110,112]. Compared to protein-coding genes, lncRNA modifications are particular to tumors. This particularity provides lncRNAs with important diagnostic biomarkers [99,121,122,123,124,125]. In HCC, some lncRNAs play important controlling roles in the growth and metastasis of HCC [126,127,128] by halting the cell cycle, preventing cell death, and enhancing DNA injury repair. LncRNAs can perform significant functions with respect to chemo- and radio resistance of tumors [129], which could be used to identify possible targets and explore novel strategies for chemo- and radiotherapy in HCC [20,28,130].

## 4. Cancer-Associated lncRNAs

In comparison to healthy controls, most lncRNAs are found in patients with malignant tumors [25,49]. Due to high expression levels of lncRNAs in tumors, lncRNAs can be found in body fluids such as blood, saliva, and plasma, suggesting that circulating lncRNAs may be employed as non-invasive tools for diagnosis of various cancers, including HCC [25,49,131,132].

A number of mechanisms involving genetic, as well as environmental, changes are involved in transforming normal cells into cancer cells, with which they share some common characteristics [133,134,135]. To alter the cell physiology and regulate cancerous development, healthy cells must introduce new capabilities. Biochemical capabilities that are gained during the multiphase production of human tumors are considered the hallmarks of cancer [135]. Maintaining proliferative signaling, escaping progression suppressors, avoiding apoptosis, initiating angiogenesis, and inducing invasion and metastasis are all examples of such alterations [136]. LncRNAs are related to nearly all cancer hallmarks [90,137]. The manipulation of diverse mechanisms is responsible for the effect of such lncRNAs in cancer hallmarks [71,135,138].

A strong relationship between tumors and lncRNAs has been identified [139]. Differential expression of lncRNAs in nearby normal and tumor tissues, as well as in normal and malignant cell lines, makes lncRNAs potential cancer biomarkers [67,125,139]. Because lncRNAs can change cell growth by modifying expressions of genes, dysregulated expressions of lncRNAs may contribute to cancer pathophysiology [74]. In such situations, the changes within lncRNAs are linked to cancer. LncRNAs can be used as potential therapeutic biomarkers [140,141,142,143,144,145,146]. Cancer-causing and anticancer lncRNAs are two types of lncRNAs in tumors (Table 1). Due to their ability to interact with molecules of DNA, protein, and RNA, as well as the ability to alter many cancer hallmarks, lncRNAs play important roles in tumor progression [73,138,141,142]. The splicing of precursor mRNAs is affected by lncRNA-mediated gene expression control, which occurs in the post-transcriptional stage. The stability of mRNAs and proteins, as well as nuclear trafficking, is factors to be considered [147,148,149,150]. More than 8000 lncRNAs have been discovered in cancer cells [151,152]. Owing to their considerable quantity and specificity of expression, such lncRNAs are effective biomarkers and strong therapeutic targets (Table 1).

## 5. LncRNAs in HCC

LncRNAs perform important functions with respect to the induction and development of HCC [69], with increased expression levels of 27 kinds of lncRNAs. Actin filamentin-1 antisense RNA (AFAP-AS1), zinc finger E-box binding homeobox 1-antisense 1 (ZEB-1-AS1), and HOX transcript antisense intergenic RNA (HOTAIR) are correlated with poor prognosis of HCC, whereas reduced expressions of 18 lncRNAs, including growth arrest-specific transcript 5 (GAS5), XIST, and maternally expressed gene 3 (MEG3), are correlated with an even worse prognosis of HCC [28]. By inducing the invasion and initiation of metastatic spread of HCC cells, lncRNA SNHG8 (small nucleolar RNA host gene 8), LINC00052, lncRNA W42 [67], LINC01225, PITPNA antisense RNA 1 (PITPNA-AS1), and ZEB1-AS1 exhibit oncogenic characteristics [120,174,175]. LncRNA-hPVT1, AFAP1-AS1, XIST, HOXA cluster antisense RNA 2 (HOXA-AS2), HOST2, and cervical carcinoma high-expressed lncRNA 1 (CCHE1) are examples of lncRNAs that can cause and promote cell proliferation, suppressing apoptosis of HCC cells [28,176]. LncRNA SNHG17 is considerably upregulated in tissues and cell lines of HCC and associated with large tumor size, poor differentiation, and the presence of vascular invasion [177]. The lncRNA TUG1-miR328-3p-SRSF9 mRNA axis works as a unique ceRNA regulator axis related to HCC malignancies [178]. LINC01194 is upregulated in the HCC cell line and controls the proliferation and migration of HCC cells by interacting with the miR-655-3p/SMAD5 axis, which provides new biomarkers for HCC diagnosis and treatment [179]. The increasing appearance of lncRNAs in HCC is assumed to be oncogenic, and lncRNAs with low expression in HCC are considered to be tumor-suppressor lncRNAs [28,99]. Those characteristics may be effective as potential therapeutic targets for HCC, especially for patients who may have already developed resistance to chemotherapeutic drugs [28,30,31,32,180]. The functions of lncRNAs in HCC are to promote cancer cell growth and invasion, repress cancer cell growth and invasion, estimate prognosis and efficacy, and act as potential biomarkers (Table 2).

The biosynthesis of lncRNAs is similar to that of protein-coding transcripts. Epigenetic modification, transcription complex recruitment, and RNA processing are all important activities that influence lncRNA production. Aberrant lncRNA biosynthesis is related to the pathogenesis of various diseases, including HCC. In comparison to non-cancerous liver tissues, high-throughput techniques such as RNA sequencing and microarray have characterized distinct lncRNA expression patterns within HCC tissues, demonstrating that lncRNA production is dysregulated throughout HCC progression [189,239]. Aberrant biogenesis activities include epigenetic activation of tumor suppression. LncRNA transcriptional repression through certain tumor-suppressive transcription factors, special processing patterns that associate lncRNAs with oncogenic activities and the binding of lncRNAs with miRNAs affect lncRNA stability [195].

### 5.1. Regulation and Modification of Chromatin by lncRNAs

Increasing evidence has shown that lncRNAs can perform a variety of functions, including epigenetic modifications in HCC [240,241] (Figure 4a). Methylation of histone and DNA is an essential epigenetic modulation that regulates gene expressions [24,242,243,244]. Inappropriate chromatin alterations of lncRNA genes, such as methylation of DNA histone modification, have generally been described throughout HCC development [245], which can cause a reduction in repressive lncRNAs of HCC and an increase in cancer-promoting lncRNAs related to HCC [195]. Linc-GALH (Gankyrin-associated LincRNA in HCC), with respect to judgment of HCC metastasis, can promote DNMT1 (DNA methyltransferase1) degradation by enthusing ubiquitination and appearance of Gankyrin (PSMD10) and decreasing HCC methylation [246]. EMT (epithelial-mesenchymal transition) is thought to be essential for tumor metastasis and relapse [247]. The up regulation of linc00441 increases H3K27 acetylation [248]. In contrast, abundantly expressed linc00441 induces DNA methyltransferases 3 alpha (DNMT3A) to methylation, deactivating the neighborhood RB1 gene to induce HCC cell proliferation [195,248]. Significantly increasing lncRNAs has been demonstrated to show their interaction with epigenetic regulator enhancer of zest homolog 2 (EZH2) to stimulate gene expression, influencing HCC metastasis [195].

### 5.2. Transcriptional Regulation and Activation

LncRNAs can regulate transcription by binding to promoters of nearby or distinct genes to recruit transcription factors to further regulate transcriptional activation [95] (Figure 4b). LncMAPK6, a mitogen-activated protein kinase 6 (MAPK6) lncRNA, has been abundantly expressed in association with liver tumor development [249]. Its interaction with RNA polymerase II recruits MAPK6 promoter, thus activating MAPK6 transcription [249]. YAP, c-Myc, and catenin are the oncogenic transcription factors that are highly expressed in HCC [250]. These transcription factors enable lncRNAs to be involved in necroptosis and cell cycle arrest in HCC, demonstrating that lncRNA transcription is critical throughout HCC [195,251].

### 5.3. Interaction with mRNAs

Certain lncRNAs affect the stabilization and translating procedures of mRNAs (Figure 4c). Through intermodulation with lncRNA-mRNA, lncRNAs activated by transforming growth factor-β (lncRNA-ATB) regulate and maximize mRNA of interleukin-11, encouraging the proliferation of circulated HCC cells in distant parts of the body [252]. Only a few primary lncRNA transcripts are affected by the exon-insertion scenario. During the formation of HCC, such process modules can play an oncogenic role. The splicing factor muscle blind-like 3 (MBNL3) was found to be overexpressed throughout the fetal liver in HCC tissues that were lacking in adults, causing LncRNA PXN-AS1 (PXN antisense RNA 1) exon 4 inclusions. Due to the splicing alteration, the lncRNA PXN-AS1 is able to interact with PXN mRNA in HCC [253]. HCC progression and metastatic spread are inhibited through lncRNA LINC01093 specific to the liver, which also acts as protein scaffolding on the way to induce insulin-like growth factor mRNA binding protein1 (IGF2BP1) and to further promote the degradation of Glioma-associated oncogene homologue 1 (GLI1) mRNA [254].

### 5.4. Sponge of MicroRNAs

LncRNAs can function as miRNA sponges, reducing deficit miRNA activity (Figure 4d). LncRNAs have been used as an extra layer of post-transcriptional regulation of gene expression [234,255]. Numerous lncRNAs have been implicated in controlling the expression of genes by interfering with miRNAs and prohibiting particular miRNAs from binding with the target mRNAs [256,257,258,259]. The HCC-associated lncRNA (HCAL) stimulates HCC metastasis by binding miR-196a, miR-196b, and miR-15a [260]. LncRNAMALAT1 (metastasis-associated lung adenocarcinoma transcription 1) can promotes migration and invasion of HCC via sponging of miR-204 [261]. In patients with HCC, a sufficient proportion of lncRNA HOXD-AS1 (HOXD cluster antisense RNA 1) expression is related to the high tumor nod [200,262]. HOXD-AS1 conservatively binds with miR-130a-3p, which can inhibit SOX4 (sex-determining region Y-related high-mobility group box transcription factor 4) to miRNA intermediated destruction, stimulating the expression of EZH2 and MMP2 to promote HCC metastasis [200]. HOXD-AS1 can also regulate the expression of Rho GTPase-activating protein 11A through highly competitive interaction with miR-19a, resulting in HCC tumor growth [262]. LncRNA HULC (highly upregulated in liver cancer) enhances HCC progression and metastasis by increasing epithelial–mesenchymal transition (EMT) progression in the miR-200a-3p/ZEB1 signaling pathway [263]. LncRNA MALAT1 enhances HCC development by sponging miR-143-3p to control ZEB1 expression [190]. MiR-34a has been reported to bind specifically with lncRNA-UFC1, causing its half-life to be reduced, thus preventing HCC growth mediated by lncRNA-UFC1 [189]. Because miRNAs are downregulated generally during HCC production, oncogenic lncRNAs are likely to be reactivated, leading to abnormal lncRNA expression profiles [195].

### 5.5. Protein Binding and/or Modification

Except for binding with miRNAs, lncRNAs are subject to biochemical processes that include protein modification (Figure 4e). Many reports have suggested that lncRNAs perform roles in protein phosphorylation modulation [264,265]. LncRNA TSLNC8 (tumor-suppressive lncRNA on chromosome 8p12) inhibits phosphorylation of STAT3 (signal transducer and activator of transcription 3) in HCC by inactivating the IL-6/STAT3 signaling pathway [264,265]. RNA-binding proteins (RBPs) have been discovered to manipulate lncRNA stabilization via physical interaction [266]. IGF2BP1/3 (insulin-like growth factor 2 mRNA-binding protein 1/3) is an RBP that binds and remains stable with long intergenic non-protein coding RNA 1138 (LINC01138) on its 220-1560-nt fragment, which is essential for HCC invasion progression [266]. Furthermore, lncRNA UFC1 can interact with another RBP, called human antigen R (HuR), via its fragment (1102-1613-nt), which is required for HCC [189]. These findings indicate that RBP-controlled lncRNA decay occurs to compensate for unusual lncRNA biogenesis in HCC.

Some reports have shown that lncRNA HNF1A-AS1 (HNF1A antisense RNA 1) prevents HCC invasion and spread by directly attaching to the C terminal of SHP-1 (SH2-containing protein tyrosine phosphatase 1), thereby stimulating phosphatase [267]. The effect of lncRNAs on the expression of genes by modification of protein is not restricted to target protein phosphorylation. In HCC, LINC01138 can exert oncogenesis behavior by interfering with arginine methyltransferases 5 (PRMT5), strengthening the stability of protein by stopping ubiquitin degradation [266]. LncRNA miR503HG comes into contact mostly with heterogeneous nuclear ribonucleoprotein A2/B1 (hnRNPA2B1) and represses metastatic tumor repression by controlling the ubiquitination status of hnRNPA2B1 [231]. By hindering CUL4A (cullin4A) intermediated ubiquitination and degradation of LATS1 (long-acting thyroid stimulator 1) within the cytoplasm, lncRNA uc.134 can disrupt HCC invasion and metastasis [224]. In particular, lncRNAs affect protein acetylation, which is an essential post-translational modification of protein control degradation [268]. Histone deacetylase 3 (HDAC3) governs lncRNA-LET (low expression in the tumor), which may be implicated in hypoxia-induced cell death [268]. LncRNA-LET inhibits Nuclear Factor 90 (NF90) protein degradation but is essential for hypoxia-induced cellular penetration [268]. LncRNAs can have a variety of effects on the formation of HCC.

### 5.6. Other Mechanisms and Pathways of lncRNAs in HCC

LncRNAs have significant effects on transcriptional, as well as post-transcriptional, regulation, relying on their subcellular localization [25]. Trans-acting nuclear lncRNAs control gene transcription epigenetically by interacting with tissue-specific chromatin modifications, such as histone-modifying complexes and DNA methyltransferases [269] (Figure 4f). Certain lncRNAs manage to sustain nuclear architecture through the scaffolding structure of the DNA-RNA-protein framework at unique sites [270,271]. Due to the genomic similarity toward their targets, the cis-acting lncRNAs may become capable of controlling gene expression inside the locus with an allele-specific method [270,271].

In cancer, lncRNAs are involved in tumor proliferation and metastasis signaling pathways [272]. The crucial mediator throughout the development of cancer is significant in HCC development and progression [273,274]. According to increasing prevalence, triggering of the catenin cascade can play a vital role in HCC [275]. Several lncRNAs play key roles in the stimulation and repression of the catenin pathway in HCC [271]. Overexpression of long intergenic non-protein-coding RNA 00210 (LINC00210) in liver tumor tissues interferes with catenin beta-interacting protein 1 (CTNNBIP1) to block the inhibitory function of CTNNBIP1 in catenin stimulation and enhance the association of catenin and TCF/LEF (T-cell factor/lymphoid enhancer factor family) complex, thereby triggering catenin signaling and liver tumor growth [276]. Some other pathways seem to be lncRNA-activated through TGF (lncRNA-ATB), further inducing EMT and aggression via highly competitive binding the miR-200 family but also modulating ZEB1 and ZEB2 [277]. In HCC, lncRNA-HEIH (HCC upregulated EZH2-associated lncRNA), in combination with enhancer of zeste homolog 2 (EZH2), performs very significant roles in G0/G1 arrest, usually requiring suppression of the EZH2 target gene [44]. Higher URHC, upregulated in HCC, can induce cell proliferation and prevent cell death by suppressing the sterile alpha motif and leucine zipper-containing kinase AZK (also known as ZAK (zipper-containing kinase) [278]. Two single nucleotide polymorphisms, rs7763881 within HULC and rs619586 within MALAT1, exist in 1344 HBV-persistent drivers and 1300 HBV-positive HCC patients [279]. Interactions of lncRNAs with some other significant signaling pathways participating in HCC metastasis and growth have been identified [280,281]. Through upregulation of PTTG1 (pituitary tumor-transforming gene 1) to trigger the PI3K/AKT signaling pathway, lncRNA PTTG3P (pituitary tumor-transforming 3 pseudo gene) enhances HCC development, as well as tumor growth [282]. HCC metastasis-promoting linc-GALH is known to be implicated in the regulation of the AKT signaling pathway [246,283]. Linc00974 also encourages the growth and migration in HCC by interfering in KRT19 (Keratin 19) [284]. LncRNA uc.134 stimulates hippo kinase signaling by preventing CUL4A from moving to the cytoplasm from the nucleus [224]. These findings illustrate that lncRNAs can function as mediating variables of the oncogenesis signaling pathways such, as Hippo kinase, Wnt, JAK/STAT, and PI3K/AKT. Although it is still unknown how lncRNAs affect HCC development, the relationship between lncRNAs and signaling pathways has paved the way for both the identification of innovative diagnostics and therapy in HCC [244,285].

## 6. Importance of Gene Expression Regulation in HCC Progression

HCC onset and development can be assessed using global genomic research due to genetic alterations that alter the expression of thousands of cancer-related genes. Hepatocarcinogenesis and the molecular pathways that underpin complicated clinical features have been studied using HCC gene regulation analysis [286,287]. The development of phenotypic expression gene profiling could revolutionize how HCC is identified and treated [286,287]. Complementary DNA microarrays for analysis of global gene expression, single-nucleotide polymorphism genotyping for identification of mutations that significantly alter gene expression and abnormal protein activities, chromosome instability mapping, and DNA–protein interactions are all widely accepted genomic data analysis technologies. In addition, several functional groups are used to develop new HCC serum diagnostic markers and therapy targets [287]. Although cancer cells disrupt EMT, it is a straightforward physiological activity that involves development and wound repair. In HCC, EMT effectors, such as fibronectin, cadherins, integrins, and vimentin, have been found to be altered, allowing for a much more mesenchymal phenotype [39,288,289,290]. In HCC, transcription factors that promote EMT, such as slug, twist, Snail, and Zeb, are upregulated [39,288,289,290]. Furthermore, the majority of studies on miRNAs, exosomes, lncRNAs, and regulatory cellular processes have been associated with EMT and found to be important in the advancement of HCC [39,288,289,290]. During primary HCC, the hypoxic microenvironment is significantly related to cancer development and angiogenesis [291]. Cancer cells interact with the aberrant microenvironment, ECM, cytokines, and chemokines and elevate the growth factors, resulting in enhanced angiogenesis [292,293]. Hepatic cells play a significant role in hepatocarcinogenesis, and the transformation of all such cells can result in cancer stem cells (CSCs) with various intrinsic factors (genetics and autoimmune diseases) and various extrinsic factors (HBV, HCV, alcohol, and AFB1), accounting for about 70–90% of the conversion of tiny hepatocyte-like progenitor cells into cancer cells [294]. Several potential surface markers of liver CSCs, such as epithelial cell adhesion molecule (EpCAM) [295], CD90 [296], CD133 [297], CD44 [298], and CD13 [299], have been identified. However, an improved understanding as to how molecular categorization and mutational confirmations influence HCC progression is required before it can be used as a targeted therapy in a medical context [296].

## 7. LncRNAs as Diagnostic and Therapeutic Markers in HCC

In HCC patients who are diagnosed later in the disease process, curative medications are no longer valuable [265]. Currently, ultrasound imaging and alpha-fetoprotein (AFP) analysis are used to diagnose HCC. Ultrasound scanning and testing are recommended in high-risk populations, and patients who undergo increasingly regular imaging have been associated with improved prognosis [300]. Nevertheless, with 47% sensitivity, surveillance imaging is insufficient to detect early-stage HCC [301]. The commonly used HCC biomarker AFP (alpha-fetoprotein) seems to have a sensitivity of 52.9%, as well as a specificity of 93.3%, which can be strengthened when combined with ultrasound imaging [302]. In the absence of HCC, some variables, including HCV infection, have also been reported to increase AFP levels [303]. However, neither ultrasound imaging nor AFP analysis reduces HCC patient mortality [304]. In early HCC, surgical procedures, including resection and liver transplantation, remain the only therapeutic choices, whereas late-stage HCC is essentially untreatable. To develop the diagnosis and treatment of HCC, new biomarkers and targeted therapies are critically required [305]. Metastasis seems to be a significant factor affecting long-term survival in patients with severe HCC [306].

### 7.1. LncRNAs as a Potential Biomarker of HCC

Patients with HCC who are diagnosed early have an increased chance of survival. Because of their tissue specificity, lncRNAs are intriguing as biomarkers [265]. It would be more appropriate to use circulating lncRNAs throughout the body fluid instead of some in malignant tissues as non-invasive markers for cancer diagnosis and surveillance [265]. However, most lncRNAs have been shown to exhibit uneven expression levels in some cancers and non-cancerous illnesses, such as cirrhosis and liver damage, resulting in diminished consistency [307]. As a result, combining lncRNAs with other chemicals, such as the well-known HCC biomarker AFP, makes a successful HCC diagnosis considerably more likely. Multiple lncRNAs, for example, UCA1 and WRAP53, in combination with AFP, ensure up to 100 percent responsiveness [307]. Similarly, combining two lncRNAs, PVT1 and uc002mbe.2, along with AFP, has been shown to serve well in the diagnosis of HCC relative to AFP alone [195,308].

As reported, lncRNA ZFAS1 (zinc finger antisense 1) is a new serum diagnostic marker for the detection of HCC [309]. The extracellular vesicle long RNAs (exLRs), which were found only in blood samples of 104 patients with HCC, can effectively distinguish HCC from non-tumor controls [310]. Consequently, a combination of serum exosomal ENSG00000258332.1 and LINC00635 with AFP is a reliable tool for HCC diagnosis [311]. LncRNA associated with micro vascular invasion in HCC (LncRNA MVIH) up regulation has been found to significantly predict persistent relapse in initial HCC patients, indicating that MVIH might be a useful marker for the early detection and individual care assessment of HCC patients [244,312,313]. The combination of XLOC014172, LINC00152, and RP11-160H22.5 could differentiate HCC patients from hepatitis patients [314]. Furthermore, the lncRNA gene polymorphism is important for HCC diagnosis [315].

### 7.2. LncRNAs as Promising Therapeutic Potentials for HCC

In addition to their potential use as diagnostic biomarkers, lncRNAs have important therapeutic techniques for new treatments of HCC [195]. The base-pairing paradigm RNA-targeting methods are simpler to implement than protein-targeting approaches. Antisense oligonucleotides (ASOs) and RNAi are the most widespread oncogenic lncRNA-targeting techniques for the treatment of HCC [316,317,318,319]. The infusion of ASOs, such as MALAT1, inhibits tumor growth in HCC-bearing nude mice [265,276,320]. ASO-mediated linc00210 absence inhibits HCC cell self-renewal and aggression, but knockout of lncRNA CASC9 (cancer susceptibility candidate 9) by RNAi decreases cancer progression in HCC [194,276]. Using precisely constructed siRNAs against lncRNAs is a technique for influencing lncRNA efficiency. The use of artificial lncRNA has been proposed to specifically target many miRNAs and may be a useful approach for resolving Sorafenib resistance in the HCC medication [321].

Discovery of more operative treatments is imperative. Recent findings have shown that a combination of atezolizumab and bevacizumab results in antitumor activity in patients with unresectable HCC [322]. *Taraxacum officinale* (L.) Weber ex F.H. Wigg, a perennial member of the Compositae family, has antitumor properties in HCC cells and has long been conventionally used as Chinese herbal medicine for liver, breast and gallbladder, hepatitis, as well as digestive, diseases [323]. According to the US Food and Drug Administration, the medication for HCC first-line therapies are bevacizumab in combination with atezolizumab, Sorafenib, and Lenvatinib; the second-line therapies include cabozantinib, pembrolizumab, ramucirumab, and regorafenib, in addition to other agents, such as bevacizumab, nivolumab, and nivolumab in combination with ipilimumab [324].

Among first-line treatments, atezolizumab in combination with bevacizumab has the highest overall survival (OS) value, although lenvatinib has the highest objective response rate (ORR) value. Among second-line treatments, cabozantinib has the highest progression-free survival (PFS) value, as well as ORR value, compared to placebo [325]. Sorafenib, the RTK-targeting drug, is perhaps the most commonly used effective medication for the treatment of HCC. LncRNA-targeting methods have certain benefits over protein-targeting approaches for the treatment of HCC [326].

Recent advancements in molecular cell biology have significantly contributed to our awareness of the molecular mechanisms of tumor genesis and its development, which, in turn, offers prospects for finding of new molecularly targeted agents to prevent molecular irregularities as promising cancer treatments [135]. Molecularly targeted treatment generally includes TKIs (tyrosine kinase inhibitors), as well as monoclonal antibodies. Five targeted therapies have been approved for treatment of progressed HCC. Among these five therapies, four are small-molecule kinase inhibitors, and the one is a monoclonal antibody against VEGFR2 (vascular endothelial growth factor receptor) [327]. In addition to the mentioned appropriate targeted therapies, various targeted therapies are in clinical trials.

Knocking out oncogenic lncRNAs and injection tumor-suppressor lncRNA may be acceptable strategies for HCC treatment. As reported, lncRNA PRAL, a tumor suppressor that acts by stabilizing p53, dramatically prevents HCC development in tumor-bearing mice [239]. ASOs and RNAi function depending on a variety of factors, such as the subcellular positioning of the target lncRNAs. ASOs perform better than RNAi in nuclei, but RNAi performs better than ASOs when it targets cytoplasmic lncRNAs [328], which may be why RNaseH is primarily found in the nucleus, although RISC is primarily found in the cytoplasm [329,330].

## 8. Future Prospects and Conclusions

Cancer-related lncRNAs are slowly but steadily becoming the most widely discussed themes, even in RNA biology, as well as oncology. According to the existing data, abnormal transcription and processing activities may result in up regulation of the tumor-promoting lncRNAs that mostly interact with DNA, RNA, and proteins. As a consequence, lncRNAs can control expression, function, and some similar characteristics of their partner binding sites, causing various cancerous phenotypes, including recurrent proliferation, irregular metabolism, and tumor growth. All of these contribute to HCC carcinogenesis and development. Given their critical functions, a subclass of lncRNAs found in body fluid may be used as HCC biomarkers, either alone or in association with other metabolites to increase specificity.

As a result, altering lncRNA expression could be a new diagnostic and treatment technique for HCC [195]. According to the US food and Drug Administration, tyrosine kinase inhibitors Sorafenib and Lenvatinib have been proven as first-line treatments, and now, bevacizumab, in combination with atezolizumab, Sorafenib, and Lenvatinib, is considered the first-line treatment for accelerated HCC [324,331].

Some lncRNAs linked to inflammatory signaling pathways, such as the IL-6/STAT3 and NF-B pathways, have been discovered. However, the exact regulatory systems that govern development from inflammation to neoplasia remain unknown. The liver is responsible for lipid metabolism and is the primary location for endogenous cholesterol metabolism. Abnormalities in these metabolic pathways promote HCC etiology, as indicated by the increased risk of HCC in patients with diabetes, extreme obesity, and hepatic steatosis [265]. Whereas significant progress has been made, the activities of lncRNAs remain unknown. LncRNAs are often questioned based on the lack of functional analyses that may be attributed to the lower sequence conservation in comparison to protein-coding genes [332]. LncRNAs prefer to sustain highly preserved secondary structures [333]. The important problem at present is thoroughly attempting to understand the main aspects of lncRNAs, such as their structures, functions, expressions, and related mechanisms. Improved statistical techniques for lncRNA biological activities can assist in identifying their significance with respect to various cancers. This knowledge may open the way for lncRNAs as potential prognostic markers and possibly even targeted therapies. In particular, strategies to target lncRNAs, including the use of siRNAs to initiate lncRNA deterioration and CRISPR/Cas9-mediated editing of the gene, must be regarded and improved. It is difficult to determine how to get the perfect molecules into appropriate cells [334]. RNA-seq is used to determine the differential expression of lncRNAs amongst tumor and non-tumor cells in an effort to explain active lncRNAs in HCC. Activities of lncRNAs are not always reflected in their variable expression patterns. Various genetic strategies are needed to illustrate lncRNA activities, which appears to be a challenging task, given thousands of lncRNAs that can only be identified simultaneously. The CRISPR sequencing technique is used to investigate the roles of protein-coding genes and lncRNAs related to screening phenotypes, proliferation, and drug resistance [335,336,337]. CRISPR analysis not only allows for identification of new functional lncRNAs that affect phenotypes of concern but makes it easier to create lncRNA-based potential therapies for a variety of human diseases [25].

## 9. Conclusions

Translating lncRNA studies into potential treatments is complex. LncRNA-based identification strategies are slowly emerging. The involvement of lncRNAs in regulation is related to the development of HCC. There are many unanswered questions at present. Future studies should concentrate on the functions and molecular pathways of lncRNAs in stimulating HCC development rather than just the concise recognition of differentially regulated and expressed lncRNAs. The main objective of gaining an improved appreciation of lncRNAs in HCC is to find new targeted therapies and biomarkers for HCC.

## Figures and Tables

**Figure 1 bioengineering-09-00406-f001:**
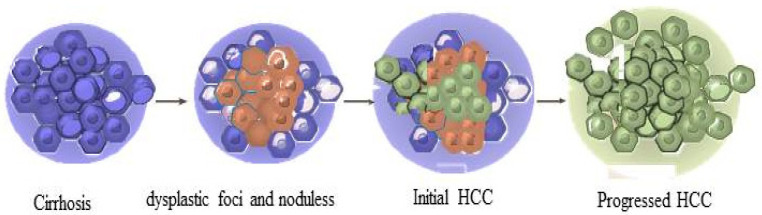
Genetic alterations in HCC. After gaining necessary genetic and epigenetic variations, cirrhosis develops into dysplastic foci and nodules to form HCC.

**Figure 2 bioengineering-09-00406-f002:**
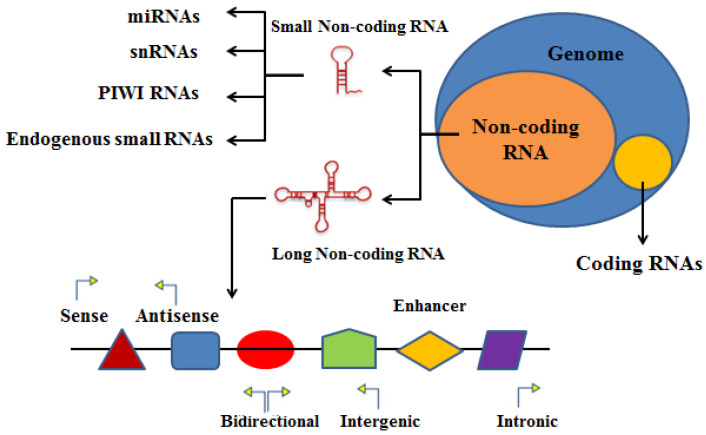
Noncoding RNAs classified into small noncoding RNAs and long noncoding RNAs. Small noncoding RNAs include miRNAs (microRNAs), snRNAs (small nucleolar RNAs), PIWI-interacting RNAs, and endogenous small interfering RNAs. Long noncoding RNAs (lncRNAs) are composed of sense, antisense, bidirectional, enhancer, intergenic, and intronic lncRNAs based on their localizations as compared to the nearby protein-coding genes. LncRNAs could function as competing endogenous RNAs (ceRNAs).

**Figure 3 bioengineering-09-00406-f003:**
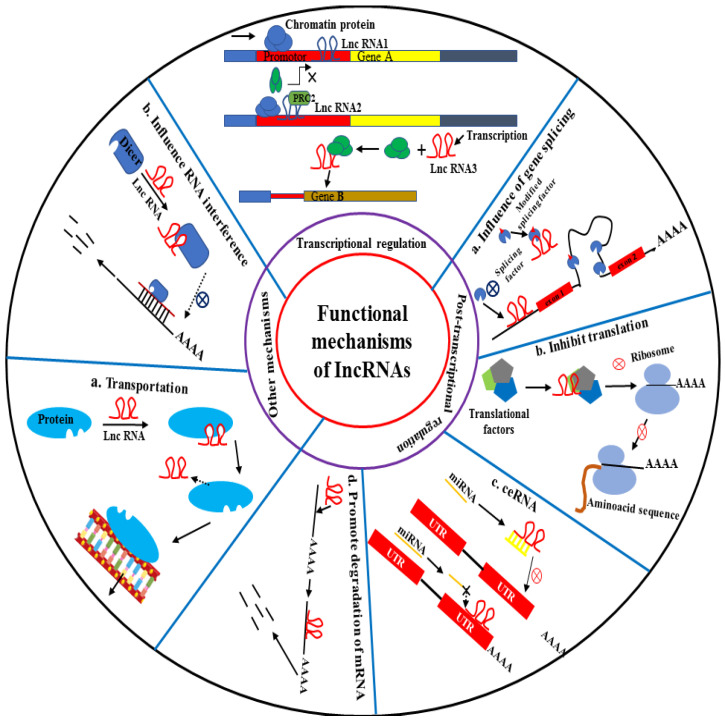
The functions of lncRNAs. LncRNAs perform a key function in gene regulation via a variety of processes, including transcriptional regulation, post-transcriptional regulation, and other mechanisms.

**Figure 4 bioengineering-09-00406-f004:**
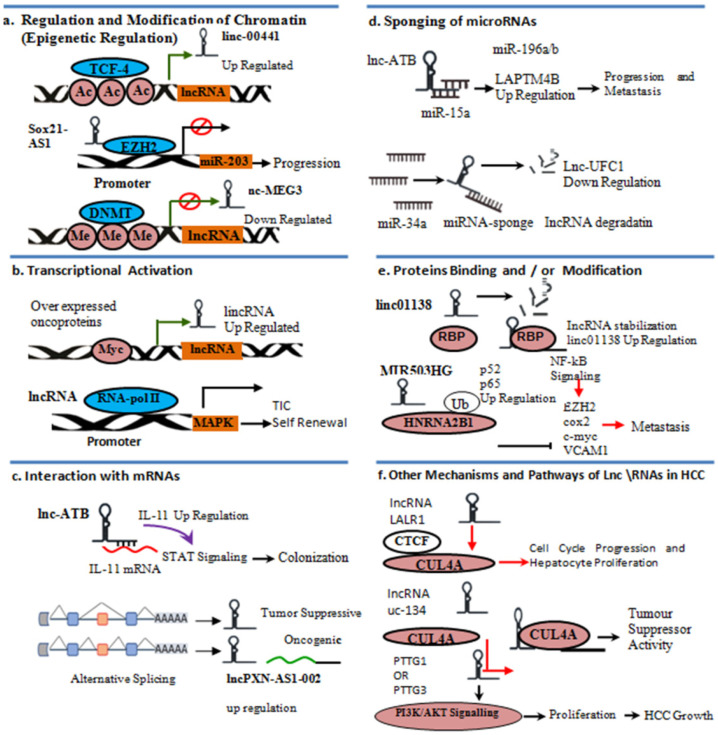
The roles of lncRNAs in HCC. (**a**) Regulation and modification of chromatin. (**b**) Transcriptional activation. (**c**) Interaction with mRNAs. (**d**) Sponging of microRNAs. (**e**) Protein binding and modification. (**f**) Other mechanisms and pathways of lncRNAs.

**Table 1 bioengineering-09-00406-t001:** Biological functions of lncRNAs in cancers.

LncRNAs	Target Pathways/ Mechanisms	Biological Functions in Cancers	Type of Cancer	Reference
LncRNA 00665	miR-224-5p/VMA21	Promoting proliferation, invasion, and migration of cancer cells	Melanoma	[146]
lncRNA RACGAP1P	miR-345-5p/RACGAP1		Breast cancer	[145]
SNHG20	miR-148a/ROCK1		Ovarian cancer	[153]
UCA1	miR-206		Cervical cancer	[154]
VCAN-AS1	p53		Gastric cancer	[155]
LINC01559	YAP		Pancreatic cancer	[156]
SNHG4	ZIC5		Prostate cancer	[157]
TTN-AS1	KLF15		Colorectal cancer	[158]
LINC00673	miR-515-5p/MARK4/Hippo		Breast cancer	[159]
RAIN	RUNX2		Breast/thyroid	[160]
PVT1	Smad3/miR-140-5p		Cervical cancer	[161]
FOXD2-AS1	miR-185-5p		Thyroid cancer	[162]
LINC00052	miR-608/EGFR		Head/neck cancer	[163]
TCONS-00020456	Smad2/PKCa	Suppression of proliferation and invasion of cancer cells	Glioblastoma cancer	[164]
ADAMTS9-AS2	CDH3		Esophageal cancer	[165]
ENST00000489676	MiR-922		Thyroid cancer	[166]
OSER1-AS1	miR-372-3p/Rab23		Hepatocellular carcinoma	[167]
HOXA-AS3	HOXA3	Prognosis and efficacy	NSCL cancer	[168]
ADAMTS9-AS2	FUS/MDM2		Glioblastoma cancer	[169]
UCA1, H19	5-fluorouracil		Rectal cancer	[170]
SNHG12	------	Potential biomarkers	Pan-cancer	[171]
HOTAIR	------		Breast cancer	[172]
SNHG11	------		Colorectal cancer	[173]

**Table 2 bioengineering-09-00406-t002:** Biological functions of lncRNAs in hepatocellular carcinoma (HCC).

LncRNAs	Target Pathways/Mechanisms	Biological Functions in HCC	Reference
LncRNA CYTOR	miR-125b/SEMA4C	Promoting proliferation, invasion, and migration of cancer cells Angiogenesis and metastasis Tumorigenesis and EMT Growth and metastasis Progression and angiogenesis	[181]
DNAJC3-AS1	miR-27b		[182]
LncRNA SNHG8	miR-542-3p and miR-4701-5p		[175]
MCM3AP-AS1	miR-194-5p/FOXA1 axis		[183]
RNA LINC00908	Sox-4		[184]
SNHG15	miR-490-3p/histone deacetylase 2 axis		[185]
GIHCG	miR-200b/a/429 PPAR gamma		[186]
ANRIL	EZH2 protein Target gene DNA		[187]
TUG1	EZH2 protein Target gene DNA		[188]
UFC1	β-catenin mRNA HuR protein		[189]
MALAT1	miR-143-3p		[190]
ICR	ICAM-1 mRNA		[191]
ZFAS1	miR-150		[192]
MVIH	PGK1 protein		[193]
CASC9	HNRNPL protein		[194,195]
LncCAMTA1	CAMTA1		[196]
Ftx	PPAR gamma		[197]
ATB	Autophagy-related protein		[198]
PDPK2P	PDK1/AKT/Caspase 3		[199]
HOXD-AS1	SOX4		[200]
HIS	ERK&AKT/GSK-3b		[201]
HOTAIR	OGFr, miR-122, SETD2		[202,203,204,205,206]
LINC00161	Activate ROCK2, miR-590-3p		[207]
DLGAP1-AS1	miR-26a/b-5p/IL-6/JAK2/STAT3		[208]
91H	IGF2		[209]
MYLK-AS1	miR-424-5p/E2F7 & activating VEGFR-2		[210]
Linc-ROR	DEPDC1		[211]
HULC	HULC/miR-383-5p/VAMP2		[212]
LINC00238	miR-522/SFRP2/DKK1	Suppression of proliferation, invasion, and migration Suppression of HCC progression	[213]
TMEM220-AS1	TMEM220/β-catenin		[214]
NBR2	JNK/ERK		[215]
lncRNA W5	---------		[216]
GAS8-AS1	GAS8		[217]
MIR22HG	miR-10a-5p/NCOR2		[218]
MIR31HG	microRNA-575		[219]
GAS5	miR182/ANGPTL1		[220]
FENDRR	miR-423-5p		[221]
EPB41L4A-AS2	miR301a-5p/FOXL1		[222]
TCONS_00006195	ENO1		[223]
Uc.134	LATS1		[224]
SVUGP2	MMP2 and 9		[225]
RP11-286H15.1	PABPC4 Ubiquitination		[226]
LncRNA-Dreh	Vimentin protein		[227]
XIST	miR-92b		[228]
LINC00221	lncRNA–miRNA–mRNA miR-485-5p/BSG miR-195-5p/MACC1 ---- HNRNPA2B1/NF-KB Caspase-8/LSD1/H3K9me3 ------ ------ miR-195/EYA1 axis	Prognosis and efficacy	[49] [229] [230] [231] [232] [204] [82] [233]
LOC554202			
LncRNA DDX11-AS1			
RP11-464I1.1			
miR503HG			
MALAT1, HOTAIR, MDG			
HOTAIR			
MIR22HG, CTC-297N7.9,			
CTD-2139B15.2, RP11-589N15.2,			
RP11-343N15.5, and			
RP11-479G22.8			
LINC00511			
lncRNA W42	DBN1 miR-448/ROCK1 ----- ----- ----- ----- -----	Potential biomarkers	[67] [120] [234] [235] [236] [237] [238]
PITPNA-AS1			
PVT1, uc002mbe.2 e			
UCA1			
RP11-486O12.2, RP11-273G15.2, RP11 863K10.7 and LINC01093			
LRB1			
ELMO1-AS1			

## Data Availability

All data generated or analyzed during this study are included in the review manuscript.

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
