# Peer review of "Function of the Long Noncoding RNAs in Hepatocellular Carcinoma: Classification, Molecular Mechanisms, and Significant Therapeutic Potentials"

_bioengineering, 2022, doi:10.3390/bioengineering9080406_

Round 1
Reviewer 1 Report
The authors aimed to summarize the role of lncRNAs in controlling the development of HCC. However, the current manuscript failed to comprehensively include lncRNAs that are associated with the pathology or progression of HCC. Many review papers about the same topic have been published. The authors need to explicitly point out what’s new in this manuscript and what’s not included. My specific comments are listed below:
1. The author failed to include literature regarding the function of lncRNAs in HCC. For example, the author mentioned PVT1, but the author did not include the paper by Wang et al (Hepatology 2014; 60(4):1278–90) that showed the function of PVT1 in HCC. This shows that the authors did not perform comprehensive search/review HCC-associated lncRNAs.
2. In my opinion, Huang et al. (2020) did a better job in summarizing the dysregulation of lncRNAs expression in HCC and their tumor suppressive or oncogenic roles during HCC tumorigenesis. The authors need to show what’s new in this manuscript and should at least include lncRNAs mentioned in Huang et al. and add more recent findings.
Huang, Z., Zhou, JK., Peng, Y. et al. The role of long noncoding RNAs in hepatocellular carcinoma. Mol Cancer 19, 77 (2020).
3. Professional English editing is required. The current version contains many grammar errors (including the title and abstract) and is not concise.
4. What does “//”in Table 1 represent? Also, the Targets path- ways/mechanisms does not line well with LncRNAs. Table 1 has to be modified and Table 2 has the same issue.
5. Resolution for Figure 3 is poor. Not acceptable for a review paper.
6. In table 2, most “Biological functions in HCC “ are “//”. The author should at least indicate whether its function is pro- or anti- tumor.
Author Response
Dear Reviewer-1,
I am pleased to know that our manuscript entitled “Function of the long noncoding RNAs in hepatocellular carcinoma: its classification, molecular mechanisms and significant therapeutic potentials” (Manuscript ID: bioengineering-1822914) has the opportunity to be revised. I would like to take this opportunity to appreciate your effects and to thank you for your valuable suggestions and comments. The manuscript has been subjected to revision accordingly. The particulars are listed below.

Reviewer 2 Report
This manuscript represents an important effort in the research field of long noncoding RNAs in primary liver tumors.
The manuscript contains 308 bibliographic references, substantially covering a large part of the research in this field. It is very difficult to be able to express an opinion on this interesting revision. In my opinion it seems complete and exhaustive.
The part of the introduction, on the epidemiology and treatment of hepatocellular carcinoma seems excessively long and could be shortened.
Regarding the medical treatment of hepatocellular carcinoma, the authors should rely on: Su GL, AGA Clinical Practice Guideline on Systemic Therapy for Hepatocellular Carcinoma. Gastroenterology. 2022 Mar; 162 (3): 920-934.
Approved therapies should include Finn RS; IMbrave150 Investigators. Atezolizumab plus Bevacizumab in Unresectable Hepatocellular Carcinoma. N Engl J Med. 2020 May 14; 382 (20): 1894-1905 which is now approved for the treatment of hepatocellular carcinoma.
Additionally, they should review Marron TU, Neoadjuvant cemiplimab for resectable hepatocellular carcinoma: a single-arm, open-label, phase 2 trial. Lancet Gastroenterol Hepatol. 2022 Mar; 7 (3): 219-229. This is because it seems that the medical therapy of primary liver cancer is leaving Sorafenib and Lenvatinib (Page 16, line 612) moving more and more towards the side of immunotherapy.
If possible, the authors should comment a little more on this aspect (including clinical).
Author Response
Dear Reviewer-2,
I am pleased to know that our manuscript entitled “Function of the long noncoding RNAs in hepatocellular carcinoma: its classification, molecular mechanisms and significant therapeutic potentials” (Manuscript ID: bioengineering-1822914) has the opportunity to be revised. I would like to take this opportunity to appreciate your effects and to thank you for your valuable suggestions and comments. The manuscript has been subjected to revision accordingly. The particulars are listed below.

Reviewer 3 Report
The human genome contains less than 1 percentage of exons transduced and translated in proteins. Although approximately 10% of the whole human genome is transcribed into RNA molecules, the large portions of RNA sequences do not code for functional proteins. In recent years, these non-coding RNAs have been intensively studied and classified according to their lengths and characteristics. Their role in the pathophysiology of human diseases has been proposed and described (Esteller M. Nat Rev Genet 2011;12:861). Khan and Zhang offered a very nice and detailed overview on long non coding RNA molecules, with specific interest in hepatocellular carcinoma and lncRNA specific role in pathogenesis and progression.
Altered expression levels of miRNAs are a hallmark in diseased conditions, and the regulation of gene expression by miRNAs plays a critical role in the pathogenesis of various human disorders, particularly chronic liver diseases. Several interventional strategies (both based on surgical and pharmacological approaches) have been proposed and translated to the clinic to ameliorate or rescue patients with liver carcinoma. The current manuscript describes some but lacks critical discussion with a glimpse of future advanced therapies.
Plenty of reports are present in scientific literature, highlighting a potential role for long non-coding RNAs in tissue-pathophysiology and development. LncRNA resistance to both RNase R and RNA exonuclease has also attracted attention for their potential role in different hepatic pathological conditions, such as steatosis or ischemic damages (Zhang P, et al. Scientific Reports 2019). And the role of lncRNAs has been proposed as important regulators of gene expression and pathological processes. Non-coding RNAs could be useful as biomarkers to diagnose liver diseases including liver fibrosis and cancer (as the authors correctly mentioned in line 253 “lncRNAs could be employed as non-invasive diagnostic indicators”). Such an important concept should be elaborated on and discussed in more detail and a list of applications compiled.
Furthermore, lncRNAs have been proposed as a novel therapeutic target to regulate specific gene expression and neoplastic pathogenesis/progression. Recent results underlined up- and down-regulation effects in different lncRNA, identified and analyzed by Gene Ontology (GO) and Kyoto Encyclopedia of Genes and Genomes (KEGG) pathway.
A critical, hypothetical network of different RNA molecules (not only lncRNA and mRNA, but also sncRNA, circRNA, miRNAand mitRNA) would be beneficial and of interest.
Furthermore, a specific discussion on lncRNA expression in the different hepatic zone (centrolobular vs pericentral) would be extremely interesting and would increment the level of novelty in a manuscript like this. In addition to HCC, lncRNAs have been reported critical for cholangiopathies and cholangiocarcinoma.
The manuscript contains several general concepts and statements. The manuscript is quite descriptive and would largely benefit from a critical revision and shortened version. Several statements and concepts are repeated multiple times and several redundancies have been found in all the sections. A concise explanation for the statement highlighting higher HCC pathogenesis in patients affected by alpha1-antitrypsin deficiency, tyrosinemia, or Wilson disease (page 2) I encouraged. Also, fatty liver diseases as poor prognosis disease in relation to HCC (page 2) should be shorty motivated
Very critical and important information, whose role and repercussion in current and advanced medical treatments should be elaborated and critically discussed. Statements as “most lncRNAs are highly selective in patients with malignant tumors” (line 250) should be avoided and replaced with more specific details
Page 2 description in particular is quite chaotic and contains several redundancies. We would require the authors to critically revise it and shorten it. Novel treatment or current pharmacological regimens are superficially and quite confusedly described. Prognostic tools and incidence should be revised and reconsidered (eventually elaborated when pertinent).
On page 2, line 92, the authors introduced the critical concepts required for the scope of the manuscript. Such an important description should be confined in paragraph 2 and properly introduced/described. A critical revision and homogenous description of transcriptomic molecules should be provided. A supporting figure or table would improve the author's description and support the reader during the reading of the forthcoming description.
On page 7 (line 276), “Cancer-causing and anti-cancer lncRNAs are two types of lncRNAs in tumors” actually introduces a very important paragraph, laced with great examples and axiomatic concepts. Maybe such a section could be elaborated and supported by a table.
But it is only on page 8 (section 5) that the authors reached the core of their work: LncRNAs in HCC. All the afore pages are introductive and may benefit from a shortened version.
Minor details:
The “Hepatocellular carcinoma” acronym should be used when not strictly requiring a full statement.
AFP in line 86 is spelled wrongly
Author Response
Dear Reviewer-3,
I am pleased to know that our manuscript entitled “Function of the long noncoding RNAs in hepatocellular carcinoma: its classification, molecular mechanisms and significant therapeutic potentials” (Manuscript ID: bioengineering-1822914) has the opportunity to be revised. I would like to take this opportunity to appreciate your effects and to thank you for your valuable suggestions and comments. The manuscript has been subjected to revision accordingly. The particulars are listed below.

Round 2
Reviewer 1 Report
The author has tried to addressed my concerns.
Author Response
I am pleased to know that our manuscript entitled “Function of the long noncoding RNAs in hepatocellular carcinoma: its classification, molecular mechanisms and significant therapeutic potentials” (Manuscript ID: bioengineering-1822914) has the opportunity to be revised. I would like to take this opportunity to appreciate your effects and to thank you for your valuable suggestions and comments. The manuscript has been subjected to revision accordingly. The particulars are listed below.

Reviewer 3 Report
Lines 31-32, sentence “Even though 31 the researchers have tried their best to conduct extensive studies to formulate more 32 successful therapeutic approaches, numerous cancers have been related to higher causes 33 of morbidity and mortality “ needs to be rephrased. Research efforts have nothing to do with morbidity/mortality.
Line 42-43: a short commentary or explanation in support of increased risks for HCC in A1AT, Wilson, or tyrosinemia would be pertinent. Is this because of mutations/missing activity?
The figure the authors mislead as supplementary in the rebuttal and later included in the Introduction section as Figure 1 is not particularly informative nor strictly required in support of the text. It should be considered as (part of) a graphic abstract maybe
The authors are mentioning and to some extent described different RNA molecules (circRNA, snRNA, PIWI RNA, endogenous sRNA, lncRNA, siRNA, miRNA at page 3 and 4, while at page 10 they introduced the ceRNA). Is ceRNA molecule a new terminology the authors introduced? Please include reference or elaborate on such concept and specify if worldwide accepted by the scientific community. However, their distinction and Figure 2, in particular, are quite confusing and misleading. Please have a final revision and eventually couple Fig.2 with a legend/table describing such molecule properties.
Lines 538-539, please revise the following text: “Taraxacum officinale (L.) Weber ex F.H.Wigg is a 538 perennial herb of Compositae, and Taraxacum officinale polysaccharide has certain 539 anti-tumor effect on HCC cells”.
Line 541: “bevacizumab+atezolizumab” should be read as “bevacizumab in combination with atezolizumab”. Similarly for all the following “+” expressions.
Line 552 “In the last eras”? what do the authors mean? I am quite confident the authors are not intentionally introducing the Geologic Time Scale. Please revise and correct such inappropriate expression.
Lines 582-584, please revise language and correct grammar mistakes
Finally, the authors keep referring to the fact that developing and translating new therapeutic strategies specifically targeting lncRNA is not an easy task and produced limited results so far. However, unfortunately, the authors did not include their expert opinion or explanation for such delay and complication in optimizing ad hoc therapies. They partially mentioned or superficially described such delay and difficulties. This is the major weakness of the excellent manuscript compiled. Sentences such as “The involvement of lncRNAs in the regulation of lncRNAs is related to the development of HCC” are extremely confused and almost non-sense to a reader. We would like to recommend a final, critical revision, with particular attention to the English language and repetitions.
Author Response

(The authors gave the same response as above.)
